# Xylitol’s Health Benefits beyond Dental Health: A Comprehensive Review

**DOI:** 10.3390/nu11081813

**Published:** 2019-08-06

**Authors:** Krista Salli, Markus J. Lehtinen, Kirsti Tiihonen, Arthur C. Ouwehand

**Affiliations:** Global Health & Nutrition Sciences, DuPont Nutrition & Biosciences, 02460 Kantvik, Finland

**Keywords:** sugar alcohol, prebiotic, bowel function, immune function, respiratory tract infections, otitis media, sinusitis, weight management, satiety, bone health

## Abstract

Xylitol has been widely documented to have dental health benefits, such as reducing the risk for dental caries. Here we report on other health benefits that have been investigated for xylitol. In skin, xylitol has been reported to improve barrier function and suppress the growth of potential skin pathogens. As a non-digestible carbohydrate, xylitol enters the colon where it is fermented by members of the colonic microbiota; species of the genus *Anaerostipes* have been reported to ferment xylitol and produce butyrate. The most common *Lactobacillus* and *Bifidobacterium* species do not appear to be able to grow on xylitol. The non-digestible but fermentable nature of xylitol also contributes to a constipation relieving effect and improved bone mineral density. Xylitol also modulates the immune system, which, together with its antimicrobial activity contribute to a reduced respiratory tract infection, sinusitis, and otitis media risk. As a low caloric sweetener, xylitol may contribute to weight management. It has been suggested that xylitol also increases satiety, but these results are not convincing yet. The benefit of xylitol on metabolic health, in addition to the benefit of the mere replacement of sucrose, remains to be determined in humans. Additional health benefits of xylitol have thus been reported and indicate further opportunities but need to be confirmed in human studies.

## 1. Introduction

Xylitol is a five-carbon sugar alcohol (C_5_H_12_O_5_, Figure 1) with a molecular weight of 152.15 g/mol, which is commonly used as a sweetener in sugar-free confectionery. It also naturally occurs in fruits and vegetables (plums, strawberries, cauliflower, and pumpkin [1]). It is equisweet to sucrose and has a very similar sweetness-time intensity to sucrose. Xylitol is the sweetest of all polyols [2]. Xylitol is best known for its dental benefits, such as reducing the risk for dental caries [3]. This is thought to function through three mechanisms: xylitol replaces cariogenic sucrose, xylitol may stimulate salivation, and xylitol may have specific inhibitory effects on *Streptococcus mutans*—the main causative microbe of dental caries [4]. Although a recent meta-analysis concluded that there is a need for high-quality studies on the dental benefits of xylitol, the same study concluded nevertheless that xylitol is an effective strategy as a self-applied caries preventive agent [3]. Furthermore, the European Food Safety Agency has approved a health claim “xylitol chewing gum reduces the risk of caries in children” [5]. Here, however, we want to focus on other potential health benefits of xylitol, such as skincare, respiratory, digestive, immune health, and weight management.

Approximately half of the consumed xylitol is absorbed; the liver readily converts it to xylose by a non-specific cytoplasmic NAD-dependent dehydrogenase. The formed xylose is phosphorylated via a specific xylulokinase to xylulose-5-phosphate, an intermediate of the pentose-phosphate pathway before conversion to glucose, which is only slowly released into the bloodstream or stored as glycogen [6,7].

Xylitol is safe for human consumption and in general well tolerated. However, as with all sugar alcohols, overconsumption (>20 g) is associated with digestive symptoms such as bloating and loose stools [8]. When consumption is seized, the symptoms disappear.

## 2. Skin

### 2.1. Skin Introduction

The skin acts as a barrier between the body and its surrounding environment. The epidermis is made up of the stratum corneum (outermost layer of the skin, Figure 2); formed by terminally differentiated epidermal keratinocytes and lipids, which play a main role as a physical and chemical permeability barrier. Under this lies the stratum granulosum, which forms a paracellular barrier that regulates the loss of moisture through the skin, as shown in Figure 2. Below that are the stratum spinosum, basal cells, and melanocytes, which are also part of the epidermis. The epidermal barrier, which is constantly being renewed, is characterized by its capacity to adapt to changing conditions in the environment [9]. The dermis, the next layer, supports the epidermis and produces matrix proteins such as elastin and collagen, as shown in Figure 2.

### 2.2. Xylitol Benefits to Skin

Xylitol (100 mM) for 2 h has been observed, in an epidermal-equivalent skin model, to improve lipid fluidity in the uppermost layer of the stratum granulosum. The model consisted of normal human epidermal keratinocytes (NHEKs); isolated from donated skin samples; cultured *ex vivo*, and studied microscopically using lipid specific staining. The improved lipid fluidity accelerated the release of lipids and accelerates the exocytosis of lamellar bodies to the intercellular domain between stratum granulosum and stratum corneum thereby improving the lamellar structure and accelerating epidermal permeability barrier recovery [10]. Indeed, volunteers (*n* = 7) who had the inside of their forearms mechanically irritated by repeated tape stripping, were observed to have significantly less moisture loss; approximately 20%, when exposed to 100 mM xylitol for 10 min as compared to water. This was measurable both 1.5 and 2 h after exposure [10].

Further studies with NHEKs have shown that the viability and intracellular calcium concentration were not affected by 0.0045%–0.45% xylitol (calcium regulates keratinocyte differentiation) after 24 and 48 h as compared to the cell culture medium alone. However, xylitol up-regulated the expression of filaggrin, loricrin, involucrin, and occludin mRNA as measured by qPCR [11]. These proteins are involved in barrier function and tight junction (TJ) formation in the skin; occludin is the major protein in TJs, filaggrin or filament aggregating protein is a filament associated protein that binds keratin fibers in epithelial cells, loricrin is the major protein in cornified cells and contributes to barrier function of the skin, involucrin is bound to loricrin [12]. Moreover, 0.45% xylitol stimulated the mitogen-activated protein kinase (MAPK) pathway in the NHEKs and induced the activation-dependent translocation of protein kinase Cδ, after 48h as determined by Western blotting, a key promoter of epidermal differentiation [11]. The effect on the other cell types in the epidermis was not investigated in this model. Twelve healthy volunteers with dry skin received topical exposure to a combination of 5% glycerol and 5% xylitol for 14 days. This was observed to be associated with increased hydration, reduced moisture loss and increased dermal and epidermal thickness, as measured from biopsies and histological staining, compared to the untreated control arm of the same volunteer. In agreement with the above-described *ex vivo* keratinocyte studies, increased expression of filaggrin in epidermal cells in biopsies taken from the volunteers was also observed [13]. The separate contribution of xylitol and glycerol in the observed effects cannot be determined from this study.

In a study with hairless mice (23/group), skin irritation induced by 3 h topical application of 5% sodium dodecyl sulfate (SDS) was reduced with concomitant exposure to 8.26% xylitol or 5% glycerol (same osmolarity); transepidermal water loss was reduced and in the irritated area blood flow was reduced as well, as determined by videomicroscopy. Histological staining indicated that the epidermal thickness was increased in response to xylitol treatment compared to SDS alone [14]. Also in healthy adult volunteers (*n* = 16), the transepidermal water loss induced by experimental irritation with 0.1% SDS could be inhibited by simultaneous exposure for 24 h to 4.5% or 15% xylitol and 2.6% or 9.0% glycerol, but not 5.4% or 18% mannitol (same osmolarity) as compared to another site on the same arm with 0.1% SDS alone for 24 h [15]. These results suggest a polyol-specific response.

In a study with male rats, the inclusion of 10% xylitol to basic chow for 20 months was observed to be associated with a thicker skin and more acid-soluble collagen was observed, as determined from biopsies. Also, less collagen fluorescence was observed, which is a marker for collagen glycosylation and aging [16]. However, no difference in collagenase soluble and insoluble collagen was observed nor more total collagen as compared to control animals fed the same chow without xylitol [17]. Three months dietary supplementation with 10% xylitol in basic chow has been reported to increase the amounts of acid-soluble and total collagen (expressed as hydroxyproline) in the skin of streptozotocin-induced type 1 diabetic male rats (10 animals/group) as compared to type 1 diabetic animals fed unsupplemented chow. Also here, reduced hexose concentrations of acid-soluble collagen and reduce fluorescence of the collagenase-soluble fraction; indicating reduced glycosylation were observed. Similar observations on increased were made for non-diabetic rats (10 animals/group) after three months on 10% xylitol supplemented chow as compared to non-diabetic rats fed unsupplemented chow; for acid-soluble and total collagen, as well as reduced hexose concentrations of acid-soluble collagen and reduced fluorescence of the collagenase-soluble fraction in the skin [18].

The selective antimicrobial activity of xylitol, observed in dental health, has also been applied to wound care. *In vitro* studies with a Lubbock Chronic Wound Biofilm model have shown that the application of 2%, 10%, and 20% xylitol in water reduced growth *Pseudomonas aeruginosa*, *Staphylococcus aureus*, and *Enterococcus faecalis* compared to the water control. The highest concentration was observed to completely abolish biofilm formation [19]. Furthermore, another *in vitro* study showed that the combination of 5% xylitol and 2% lactoferrin could reduce the biofilm formation of *P. aeruginosa* and methicillin-resistant *S. aureus* after 72 h in a colony drip flow reactor, as compared to base wound dressing alone [20]. The anti-*S. aureus* potential of xylitol has also been investigated in human volunteers. Seventeen volunteers with atopic dermatitis received skin lotion with or without a combination of 5% xylitol and 0.2% farnesol on either arm for seven days. Compared to the control arm treated with unsupplemented lotion, *S. aureus* was significantly reduced, and skin moisture increased [21]. The contribution of xylitol alone cannot be deduced from this study. A further potential benefit of xylitol in wound care is the negative dissolution energy [2] which gives a cooling effect to the tissue.

### 2.3. Conclusions

Topical exposure of the skin with xylitol has thus been shown to reduce skin moisture loss. The mechanism appears to relate to increased tight junction and barrier formation in the skin. Also, dietary exposure to xylitol has been found to improve skin thickness. The antimicrobial activity against skin pathogens has been documented mainly in combination with other compounds and the contribution of xylitol to the observed effects needs to be determined. Furthermore, many of these results have been obtained *in vitro* and in animal models at relatively high doses (10% of the diet); their applicability to humans thus still needs to confirmed.

## 3. Digestive Tract

### 3.1. Introduction

The digestive tract can be largely divided into the stomach, small intestine, and large intestine (colon). Much of the digestion and nutrient absorption takes place in the stomach and small intestine. Although the upper digestive tract harbors a microbiota [22], it is especially the colon that is host to a diverse and extensive microbiota [23]. This colonic microbiota ferments non-digested dietary components, mainly fiber, and other components that have escaped digestion as well as sloughed-off cells and secretions. The colon absorbs the fermentation products together with water from the digesta; in particular short-chain fatty acids are an important additional energy source.

Xylitol is not digested by human enzymes and approximately 50% of the consumed xylitol is absorbed through passive diffusion in the small intestine [6]. The remaining 50% of the dietary xylitol thus enters the colon where it can serve as an energy and carbon source for the intestinal microbiota and leads to the formation of short-chain fatty acids which provide energy to the host and support immune system homeostasis [24]. These properties of xylitol are very similar to what is expected from a prebiotic; a substrate that is selectively utilized by host microorganisms conferring a health benefit [25]. The increased concentration of xylitol in the digesta leads to an increased osmotic pressure which contributes to water retention in the digesta and thus may lead to laxative effects when consumed in excess (>20 g) [8,24]. However, this property of xylitol can also be used to address constipation; which is in line with the prebiotic nature of xylitol.

### 3.2. Prebiotic Benefits of Xylitol

Simulations of fermentation by the colonic microbiota *in vitro* have shown that exposure of this microbiota to xylitol leads to a rapid disappearance of the xylitol, as determined by enzymatic colorimetry, indicating that it is readily fermented by the simulated intestinal microbiota. Gas chromatographic analysis of the simulated colonic digesta showed an increased formation of butyric acid compared to the non-supplemented control simulations [26]. Strains from the genus *Anaerostipes* have been observed by 16S rRNA denaturing gradient gel electrophoresis (DGGE) analyses to be associated with the increased production of butyric acid in fecal cultures [27]. Production of butyric acid is considered beneficial for colonic health as it is the preferred energy source for colonocytes and is thought to be associated with a reduced risk for colorectal cancer [28]. Furthermore, butyric acid promotes the generation of regulatory T-cells that promote immune system balance [29]. In rats (at least 5 animals/group), early fecal microscopy studies indicated that 20% of dietary xylitol caused a shift from fecal Gram-negative to Gram-positive bacteria after six weeks compared to animals fed an unsupplemented diet; the magnitude of this change was, however, not reported. Similar observations were made in humans; six volunteers, after an overnight fast, consumed in a cross-over design randomly a single 30 g dose of xylitol or a single 30 g dose of glucose (control) in 200 mL water. Fecal microscopy indicated an increase in Gram-positive bacteria from 20%–30% to 50%–55% for glucose and xylitol, respectively, and a concomitant decrease in Gram-negative bacteria was observed. Furthermore, a reduction in the fecal level of yeasts was reported, from Log10 9.2–9.4 colony forming units (CFU)/g feces during the control phase to Log10 7.2–7.5 CFU/g feces after xylitol consumption [30]. The type of yeast that was reduced was not reported, but *in vitro* studies have reported that xylitol can suppress the growth of *Candida* with a minimal inhibitory concentration of 200 mg/mL and a 99.95% reduction in colony-forming units at 400 mg/mL [31]. Recent mouse studies (5 animals/group) have reported that consumption of xylitol (40 or 194 mg/kg body weight/day) for 15 weeks was associated with an increase in the genus *Prevotella*, the phyla Eubacteria and Firmicutes and a reduction in the phylum Bacteroidetes by DGGE analysis [32]. Others have made similar observations, terminal restriction fragment length polymorphism (TRFLP) analysis indicated reduced levels of *Bacteroides* and *Clostridium* cluster XIVa and increased levels of *Prevotella* in mice (7 animals/group) fed 5% xylitol for 28 days as compared to animals fed unsupplemented chow [33]. In studies with cyclophosphamide-immune suppressed mice, 5%–10% xylitol (12 animals) was observed to lead to significantly lower fecal counts of *Candida albicans* (7.58 vs. 5.22 Log10 CFU/g, control and xylitol respectively) and significantly less and fewer cases of *C. albicans* invasion of the gastric wall as compared to animals not fed xylitol (10 animals); 80% vs. 10% of animals, control and xylitol respectively [34]. Furthermore, urinary HPLC analysis indicated an increased metabolism of daidzein to equol when mouse diet (7 animals/group) was supplemented with 0.05% daidzein (control) or 0.05% daidzein and 5% xylitol for 28 days [33]; this may contribute improved bone health.

These observations are in agreement with the definition of prebiotics [25]; furthermore, xylitol is utilized only by a limited number of organisms and changes the metabolism of the microbiota; as expected for a prebiotic, Table 1. As Table 1 also clearly shows, commercial probiotics have been shown to be unable to grow on xylitol as sole carbon and energy source.

Even though organisms may not be able to metabolize and grow on xylitol, there may still be an opportunity for synergy with xylitol and probiotic bacteria, as was shown with the combination of *Lactobacillus plantarum* Inducia in combination with 5% xylitol which was reported to completely stop spore germination of *Clostridioides* (formerly *Clostridium*) *difficile*, *in vitro* after 48 h. In addition, prefeeding with a single dose of 0.2 g xylitol improved the survival of hamsters in a *C. difficile* challenge model (5 out of 9 survived in the xylitol test against 2 out of 15 in the unsupplemented group). Fecal colonization with *C. difficile* quantified by real-time PCR was lower in the xylitol group, 3.5 vs. 4.9 Log10 gene copy number/g in the control group. Real-time PCR *Lactobacillus* fecal counts, however, were highest in the xylitol group, 6.6 vs. 4.6 Log10 gene copy number/g in the control group [39].

### 3.3. Benefits of Xylitol on Bowel Function

Similar as other prebiotics [40], xylitol has been used to relieve constipation. To investigate the normalization of bowel function post-laparoscopic surgery; 60 patients were randomized to consume xylitol chewing gum (amount not reported) three times per day and 60 patients allocated to a non-chewing gum control group. The time to first flatus (−5.7 h) and first bowel sounds (3.8 h) was observed to be significantly reduced compared to the control group. There was, however, no influence on time to first bowel movement [41]. This result is very similar to what was observed with xylitol chewing gum (2.40–2.74 g xylitol/dose) every two hours until first flatus, for normalizing bowel function after Caesarian section; 40 women in xylitol chewing gum group and 40 women in non-xylitol chewing gum control group. Time to first bowel sounds (−1.1 h) and first flatus (−0.9 h) were significantly reduced, but no effect was observed for time to first bowel movement compared to the control group [42]. However, xylitol chewing gum (0.86 g xylitol/dose; 43 subjects) three times/day has been shown to contribute to earlier normalization of bowel function after elective proctectomy; time to first flatus (−6.9 h) and time to first stool (−12.3 h) were significantly reduced compared to the control group (no chewing gum; 46 subjects). Interestingly, also post-operative opioid use was reduced in the xylitol chewing group by approximately 20% as compared to the control group. No differences in post-operative complications were observed [43].

### 3.4. Conclusions

Xylitol has been shown to modulate intestinal microbial composition and activity *in vitro* and in animal studies. Although these data are promising, data in humans are limited. Similarly, for improving bowel function, human data exists but are limited to specific patient groups. There is thus a need for studies in, otherwise healthy, humans with constipation.

## 4. Nose, Throat and Ear

### 4.1. Introduction

As all the body sites that are exposed to the outside environment, also the respiratory tract is colonized by a microbiota. An important function of this microbiota is to hamper the establishment of exogenous microbes; in particular potential pathogens. As with the microbiota in other body sites, the respiratory microbiota evolves from birth to an ‘adult-like’ microbiota [44]. In contrast to viral gastrointestinal infections, it seems that during an upper respiratory tract viral infection the nasal microbiota is relatively stable as was demonstrated in an experimental rhinovirus challenge study in humans [45]. The microbiota composition also differs at different sites along the respiratory tract. The anterior nares may be colonized by *Staphylococcus* spp., *Cutibacterium* (formerly *Propionibacterium*) spp., *Streptococcus* spp. and *Corynebacterium* spp. [46]. The nasopharyngeal microbiota demonstrates considerable overlap with the anterior nares and consists of *Moraxella* spp., *Staphylococcus* spp., *Corynebacterium* spp., *Dolosigranulum* spp., *Haemophilus* spp. and *Streptococcus* spp. [46]. The microbiota of the oropharynx is characterized by *Streptococcus* spp., *Neisseria* spp., *Rothia* spp., *Veillonella* spp., *Prevotella* spp. and *Leptotrichia* spp. [46]. Some of these potential pathogens can spread from the nasopharynx into the sinus cavity during viral respiratory infection and cause sinus infection; *S. aureus*, *Staphylococcus epidermidis*, and Gram-negative bacteria such as *P. aeruginosa* and *Klebsiella pneumoniae*, predominate in chronic rhinosinusitis [47]. Acute otitis media (AOM) is defined as the presence of middle ear effusion (thick or sticky fluid behind the eardrum in the middle ear) and a rapid onset of signs or symptoms of middle-ear inflammation, such as ear pain, discharge from the ear or fever. Also here, the key step in the pathogenesis is the colonization of the upper airways with pathogenic bacteria; in particular *S. pneumoniae* and *H. influenzae*, which move from the nasopharynx through the eustachian tube to the middle ear [48].

### 4.2. Benefits of Xylitol in Respiratory Health

*In vitro* studies have shown that 1% and 5% xylitol markedly reduced the growth of alpha-hemolytic streptococci, including *S. pneumoniae* in a dose dependent manner. The inhibitory growth pattern was similar to that previously seen with *S. mutans*. Xylitol reduced slightly the growth of beta-hemolytic streptococci but not that of *H. influenzae* or *Moraxella catarrhalis* [49]. Although *in vitro* inhibition of *S. pneumoniae* was observed, nasal infection of rats (20 animals/group) with *S. pneumoniae* could not be reduced, as evaluated by PCR, with 3 day exposure to dietary xylitol (20%) or nasal spray with 5% xylitol compared to control animals not exposed to xylitol [50].

Furthermore, 250 µl of 5% xylitol sprayed for 4 days into each nostril of 21 healthy volunteers significantly decreased the number of nasal coagulase-negative *Staphylococcus* compared with saline control treatment in the same volunteers. Counts were reduced from 597 CFU/nasal swab during the control treatment to 99 CFU/nasal swab during the xylitol treatment; no other organisms were assessed [51].

A nasal spray with xylitol has been reported to improve the quality of life in patients with non-allergic nasal congestion. Subjects were randomized to either receive xylitol spray twice daily for 5 days (*n* = 14) or saline (*n* = 14). Objective rhinometry measures were not significantly different from control and baseline, and subjective measures of nasal obstruction, by questionnaire, only exhibited a trend for improvement from baseline. However, the Rhinoconjunctivitis Quality of Life Questionnaire indicated a significant improvement from baseline for the xylitol group, but not for the control group [52].

Despite some anti-pathogenic effects by xylitol on some potential pathogens of the upper respiratory tract, the consumption of 5 pieces of 15% xylitol-containing chewing gum by 106 pharyngitis patients for three months was not found to be associated with a reduction in pharyngitis and did not perform better in reducing symptoms; difficulty in swallowing and sore throat as compared to no chewing gum control subjects (*n* = 110). Data were collected by questionnaire [53]. Inhalation of xylitol aerosol has been suggested to reduce salt concentration in airway surface liquid (ASL); increased salt concentrations are associated with reduced antimicrobial activity of ASL and may partially explain the pathogenesis of cystic fibrosis [51].

As will be discussed below under immune-modulatory effects of xylitol (Section 6.2) there is substantial animal model data indicating a benefit of xylitol consumption and immune modulation which improves resistance to experimental viral infections by the human respiratory syncytial virus (hRSV) and influenza A virus (H1N1).

### 4.3. Benefits of Xylitol in Sinusitis

A reduction of the ionic composition of ASL by xylitol has been hypothesized to be beneficial not only for respiratory tract infections but also for the treatment of sinusitis. *In vitro*, 5% and 10% xylitol in saline significantly reduced *S. epidermidis* and *S. aureus* biofilm formation after 1 h, and after 24 h also of *P. aeruginosa* compared to saline. After 4 h 5% and 10% xylitol significantly reduced the growth of planktonic *S. epidermidis*, *S. aureus*, and *P. aeruginosa* compared to saline. There was no difference between 5% and 10% xylitol [54]. As mentioned above, 2%, 10%, and 20% xylitol in water have also been shown to inhibit the growth of *P. aeruginosa* in a biofilm model [19].

Indeed, in experimental sinusitis through *P. aeruginosa* infection of 26 rabbits, and local pre-administration (20 min) of 0.1 mL 5% xylitol for five days, reduced the number of recovered *P. aeruginosa* compared to administration with saline in the other sinus of the same rabbit (control). Culturing showed counts of 5.37 × 10^6^ CFU in control sinuses and 1.93 × 10^6^ CFU in xylitol pretreated sinuses. However, simultaneous or subsequent administration of xylitol and *P. aeruginosa* infection resulted only in a non-significant reduction in *P. aeruginosa* [55].

A 10-day nasal irrigation with a 5% xylitol solution by 15 subjects with chronic rhinosinusitis resulted in a significant reduction in Sino-Nasal Outcome Test 20 (SNOT-20) score compared to control irrigation with saline. The volunteers, however, did not self-report an improvement in their sino-nasal wellbeing. No adverse events were reported [56]. In a subsequent study with 30 patients with chronic rhinosinusitis, nasal irrigation with a 5% xylitol solution for 30 days has indeed been found to lead to an improvement in symptoms of chronic rhinosinusitis reported as SNOT-22 [57]. As a potential mechanism, a reduction in the viscoelasticity of mucus has been proposed [58].

### 4.4. Acute Otitis Media

As noted above, *S. pneumoniae* is one of the main causative agents of AOM; 1% and 5% xylitol has been shown to inhibit the growth of *S. pneumoniae in vitro* [49]. Ultrastructural analysis of the pneumococci showed that the cell wall became more diffuse, the polysaccharide capsule became ragged and the proportion of damaged pneumococci increased after exposure to 5% xylitol for 2 h, but not after exposure to other sugars or control medium [59]. In fact, exposure to 5% xylitol lowered pneumococcal capsular locus (*cpsB*) gene expression levels significantly compared with those in the control and glucose media [60]. However, in clinical trials, xylitol did not decrease nasopharyngeal carriage of pneumococci; even though AOM risk was reduced. Nevertheless, xylitol at 0.5% solution has been observed to reduce the growth of 20 pneumococcal clinical isolates *in vitro* compared to other carbon sources. Also *in vitro* pneumococcal biofilm formation was reduced and expression of genes involved in biofilm formation—capsule, competence, and autolysin—was reduced [61].

A recent Cochrane review investigated the benefit of the prophylactic administration of xylitol to healthy children up to 12 years of age on the risk for the development of AOM. In all, 5 clinical trials were identified and included in the analysis, which involved 3405 children in total. Doses used ranged from 8.4 to 10 g/day. The authors concluded that there is moderate-quality evidence that xylitol (in any form) can reduce the risk of AOM from 30% in the control group to approximately 22%. However, xylitol was not found to be effective in reducing AOM among healthy children during respiratory infection or among otitis-prone healthy children [48]. Furthermore, the authors expressed the concern that there is only a limited number of studies, mainly from the same research group. In that sense, it is interesting to see that at least two clinical trials are on the way to investigate the effect of xylitol on AOM (clinicaltrials.gov: NCT02950311 and NCT03055091 [62]).

### 4.5. Conclusions

Some subjective benefits for xylitol were observed in relieving congestion; overall these results are not convincing. Also for sinusitis, results are inconclusive. For AOM, however, there is quite convincing evidence on the potential benefit of xylitol in reducing its risk.

## 5. Bone

### 5.1. Introduction

Although bone may appear to be a rather static tissue, it is actually in continuous turnover. It is, therefore, important that there is a correct balance in the resorption and reconstruction of bone tissue. There is a continued risk for reduced reconstruction and especially with aging a risk for osteoporosis. Dietary means to improve mineral absorption, bone mineral density, and bone strength are thus welcome.

### 5.2. Effects of Xylitol on Bone Strength

In non-challenged animals (12 rats/group) on a diet supplemented with 10% or 20% (w/w) xylitol for 40 days, higher levels of both serum Ca^2+^ (double and triple that of the control group for 10% and 20% xylitol, respectively) and 25% and 80% increase in alkaline phosphatase activity (for 10% and 20% xylitol, respectively) were observed compared to the unsupplemented control group. Microfocus X-ray computed tomography did not show significant differences in the three-dimensional bone structure or trabecular bone structure of the femur. However, the histological analysis indicated an increase in trabeculae. Furthermore, both xylitol groups showed 3% and 6% higher bone density for 10% and 20% xylitol, respectively, than the control group fed an unsupplemented diet [63]. Xylitol has also been shown to reduce bone resorption by 42% in tetracyclin-challenged animals (10 rats/group) on a diet supplemented with 1 molar xylitol per kilogram dry feed for 31 days, compared to the control animals on a non-supplemented basal diet [64]. A similar study with 5%, 10% and 20% dietary xylitol in tetracyclin-challenged animals (10 rats/group) for 31 days noted a retarding effect on bone resorption of about 25% in the 10% xylitol group, about 40% in the 20% xylitol group, and undetectable in the 5% xylitol group. Furthermore, the effect was detected as early as 2 days after the beginning of xylitol-feeding and was maintained throughout the experimental period of 31 days compared to the unsupplemented control group [65]. This is in an agreement with observations in an ovariectomized rat model (10 animals/group). After three months on a 10% (w/w) xylitol diet, humeral ash, calcium and phosphorus loss was abrogated as compared to animals not supplemented with xylitol and no significant difference compared to sham operated animals. Furthermore, there was no loss of stress and strain resistance upon xylitol supplementation compared to sham operated animals; while elasticity was maintained. Diets between the groups were isocaloric [66].

In an injected type II collagen-induced arthritis model with 20 rats/ group, administration of 10% dietary xylitol for 17 days led to a significant protective effect against the imbalance in bone metabolism. This was seen in greater values of osteoid thickness, as well as in lower values of the number of osteoclasts on bone surface, trabecular separation, and eroded surface/bone surface in the xylitol-fed animals as compared to arthritic animals few the unsupplemented diet. In the case of trabecular bone volume, trabecular number and trabecular separation this was not different from the non-arthritic rats [67]. These observations can partially be explained by an increased bone formation activity induced by xylitol and a diminished bone resorption activity. Also, in a streptozotocin-induced type I diabetic osteoporosis model with ten rats/group, 3-month dietary supplementation with 10% and 20% xylitol has been shown to reduce the loss of trabecular bone volume and bone strength. Tibia density and ash weight in both xylitol groups were significantly different from diabetic rats fed the unsupplemented diet but similar to unsupplemented healthy rats. This was similar for tibia and femur stress tolerance and for histomorphometric assessed tibia trabecular bone volume; both xylitol groups were significantly different from diabetic rats fed the unsupplemented diet but similar to unsupplemented healthy rats [68].

As discussed above, in a mouse study, 28 days of 5% dietary xylitol was observed to stimulate the conversion of daidzian to equol [33]. The conversion of isoflavones to equol has been suggested to be responsible for their positive effects on bone health [69], whether dietary xylitol plus isoflavonoids exert a favorable effect on bone health remains, however, to be studied [33].

### 5.3. Conclusions

The ability of xylitol to positively influence bone health is in line with its prebiotic properties. Being undigestible but fermented in the colon, leads to a production of short-chain fatty acids and a reduction in pH of the digesta. This improves the solubility and absorption of minerals such as calcium. Furthermore, it has been shown in mice that butyrate stimulates bone formation via regulatory T cell-dependent mechanisms [70] thus linking the butyrogenic effect of xylitol [18] to bone health. These observations are, however, all in animals. Human studies are required to validate these benefits. Furthermore, the levels of dietary xylitol in animal studies are high (up to 20%) and not feasible for humans.

## 6. Immune Function

### 6.1. Introduction

As the first line of defense against foreign compounds and potential pathogenic micro-organisms, the body has physicochemical barriers such as the skin and mucous membranes. As mentioned above, xylitol may beneficially affect the skin barrier function, and as will be discussed below, xylitol also improves mucous membrane function; especially in the oropharynx. Below these barriers, the body relies on the immune system which can roughly be divided into a non-specific, fast-working, innate immunity and highly specific, but slower reacting, acquired immunity [71]. Xylitol may exert its effects on the immune system indirectly by prebiotic effect as discussed above or directly by influencing host (e.g., immune) cell metabolism [72].

### 6.2. Immune Modulatory Effects of Xylitol

Xylitol has been found to potentiate immune responses mainly in animal models. A single 0.5 mL dose of 20% xylitol within 24 h after hatching of ten female broiler chicks was found to improve splenocyte proliferation by B-cell and T-cell mitogens (concanavalin A and pokeweed mitogen) compared to 0.5 mL of 20% glucose. Furthermore, antibody titers to keyhole limpet hemocyanin (KHL) and *Mycobacterium butyricum* injected at day 5 were higher at day 12 post-hatching compared to animals that received glucose [73]; indicating an improved acquired immune response development in chicks. The effect of xylitol on innate immunity has been studied in rats. Rats (20 animals/group) fed 20% dietary xylitol exhibited a 6.7% higher increase in the percentage of activated neutrophils from baseline than in the unsupplemented control group after 2 weeks. Likewise, the strength of the oxidative burst per neutrophil was 13.5% higher in the xylitol group as compared to the control group [74]. When rats (20 animals/group) were infected with an intraperitoneal inoculation of *Streptococcus pneumoniae* after two weeks supplementation with 10% or 20% dietary xylitol, or no supplementation (control). The mean survival time was 11 h longer in the 10% xylitol and 12 h longer in the 20% group compared to the control group [74].

Anti-bacterial effects of xylitol have been well documented especially against oral [75] and respiratory pathogens [19]; see also earlier sections. However, only a few studies have investigated its effect on viral infections. Human respiratory syncytial virus (hRSV) is the most common cause of bronchiolitis and pneumonia in infants. There is a need for prophylactic and therapeutic strategies to control hRSV infection. Mice (5/group) receiving dietary xylitol (3.3–33 mg/kg/d in phosphate-buffered saline; PBS) for 14 d prior to hRSV challenge and for a further 3 d post-challenge had significantly lower lung virus titers compared to PBS only, control mice. In line with lower viral load, also fewer CD3(+) and CD3(+)CD8(+) lymphocytes were found in bronchoalveolar lavage, indicating less need for lymphocyte recruitment to control the viral infection [76]. Similar effects were observed for the anti-viral drug ribavirin (40 mg/kg/d during the 3 days post hRSV infection) in the study [76]. The results indicate an improved innate immune response but nevertheless combined with a reduced inflammatory response to hRSV infection. Another mouse study (five mice/group) investigated the effect of xylitol consumption (3.3 or 33 mg/kg/d) during 5 days prior to influenza infection and three days post-infection. Mortality in mice infected with influenza A virus (H1N1) could not be influenced by prophylactic oral application of xylitol or red ginseng. However, combining the two remarkably reduced mortality. With a higher dose of xylitol (33 mg/kg body weight/day) being more effective than the lower xylitol dose (3.3 mg/kg body weight/day). Interestingly, dietary administration of 33 mg/kg/d xylitol significantly reduced the lung viral titer compared the PBS control [77].

### 6.3. Anti-Inflammatory Effects of Xylitol

The studies discussed above indicate that xylitol may have anti-inflammatory effects on skin by improving the epithelial tight junctions and thus limiting the leakage of microbial and other foreign components into the host. It has been further shown that 0.0045%–0.45% xylitol exerts direct anti-inflammatory effects after 24 and 48h on NHEKs stimulated *ex vivo* with toll-like receptor agonists lipopolysaccharide (LPS), lipoteichoic acid and polyl:C, as compared to the cell culture medium alone [11]. Although the authors noted some skin donor-dependent effect, xylitol was in general effective in suppressing inflammatory cytokine interleukin (IL)-1α and IL-1β upregulation, and also in decreasing tumor necrosis factor (TNF)-α after polyl:C induction. It can be hypothesized that this reduced inflammatory response contributes to improved skin barrier function. Further evidence on anti-inflammatory effects was observed in a hairless mouse model (23 animals/group). Inflammatory responses induced by irritation of the skin for 3h with 5% SDS were substantially reduced by concomitant topical xylitol administration (at 8.26% or 16.52%); normalizing the level of lymphocytes and reducing the expression of inflammatory cytokines IL-1β and TNF-α, but not IL-1α in skin biopsies, compared to biopsies only treated with 5% SDS [14]. On the other hand, intraperitoneal injection of *Escherichia coli* LPS caused an increase in α1 acid glycoprotein in 10 and 12-day-old male broiler chicks (16 animals/group) as expected. This acute-phase inflammatory marker protein was, however, not affected by the inclusion of 6% xylitol (+9% glucose) in the diet for 7 days [78]. Nevertheless, the LPS induced reduction in body weight gain, feed intake and feed efficiency were partially prevented by the xylitol diet as compared to the 15% dietary glucose control; suggesting a reduced physiological stress response to the immune challenge. 

### 6.4. Conclusions

In animal models, xylitol has been observed to stimulate innate and acquired immunity; mainly against bacterial infectious agents. For viral infections, results are less conclusive. Also, the anti-inflammatory effects of xylitol are somewhat inconclusive and based on animal studies. Information on the potential effects on human inflammatory responses is lacking.

## 7. Weight Management

### 7.1. Introduction

Overweight and obesity are an increasing health risk not only in affluent countries but increasingly also in developing countries. Strategies to aid consumers with weight management are thus very welcome and xylitol may play a role here. A potential mechanism by which xylitol could contribute to weight management and reduced energy intake is through the induction of satiety. In addition to weight management, there may also be a benefit in counteracting the consequences of overweight and obesity, commonly referred to as metabolic syndrome; insulin resistance, high serum cholesterol and hyperlipidemia [79].

### 7.2. Effects of Xylitol on Weight Management

An obvious contribution of xylitol to weight management is through the replacement of sucrose. The caloric value of sucrose is 3.87 kcal/g and for xylitol approximately 2.4 kcal/g [2]. As xylitol is equisweet to sucrose, replacing sucrose with xylitol will reduce the caloric value of a particular food while maintaining taste. In confectionery, xylitol will also contribute the same bulk as sucrose. Whether this will contribute to long-term weight loss is uncertain.

For short-term weight management, a high-fat diet animal model (6 rats/group), reported a smaller bodyweight increase after an 8-week intervention, with less visceral (−12.9% and −15.5%) and epididymal fat (−15.5% and −17%) was observed in rats on 1 and 2 g xylitol/100 kcal of diet, respectively, as compared to animals fed an unsupplemented high-fat diet. This may be explained by the observation that adipose tissue of the xylitol-fed rats exhibited significantly higher levels of mRNAs encoding peroxisome proliferator-activated receptor (PPAR)γ, adiponectin, hormone-sensitive lipase (HSL) and adipose triglyceride lipase (ATGL). These factors regulate lipid metabolism and storage and may have caused a miniaturization of adipocytes, lipolysis, and liver fatty acid oxidation [7]. Further animal studies (12 rats/group) have also reported lower weight for animals consuming 10% or 20% dietary xylitol for 40 days; with 10% xylitol approximately 5% lower body weight and with 20% approximately 15% lower body weight [63]. In a fructose-streptozotocin-induced type 2 diabetic rat model, 7 animals/group, were fed 0 (control), 2.5%, 5% and 10% dietary xylitol for 4 weeks. A dose-dependent reduction in food and fluid intake was noted compared to diabetic control animals, where 10% xylitol was not different from non-diabetic animals. Bodyweight gain was, however, similar to the control animals but less than the healthy animals [80]. 

A one-year study with 91 obese subjects suggests an inverse relation between xylitol consumption and weight loss; a high intake of xylitol would predict for a small weight loss. People in the two lower quartiles had a 5.5-fold greater chance of losing more than 10% weight, while subjects in the highest quartile and a 14 chance of losing less than 10% weight [81]. Whether this is just a correlation or an actual causality remains to be determined.

### 7.3. Benefits of Xylitol on Satiety

Nasogastric administration of 50 g xylitol in 300 mL water to 10 obese and 10 lean volunteers after an 8 h fasting, induced an increase in cholecystokinin (CCK) and glucagon-like peptide-1 (GLP-1) compared to water alone [82]; both are indicated as satiation hormones. This was associated with an increased time to gastric emptying in both groups as compared to the control (water). However, subjective feelings of appetite were not influenced compared to the water control [82]. Similarly, an earlier study indicated that 25 g xylitol in yogurt for 10 days had no influence on reported fullness in 16 healthy lean adults. However, the combination of 12.5 g xylitol and 12.5 g polydextrose resulted in an increased subjective feeling of fullness [83]. Interestingly, clinical studies have reported that a single dose of 30 g xylitol in 200 mL water resulted in a change in gastric emptying half-time from 39.8 min during the glucose control to 77.5 min during the xylitol test with 5 healthy volunteers in a cross over design study. This delay in gastric emptying was associated with increased plasma motilin [84]. Motilin is involved in the regulation of small intestinal motility [85]. After ingestion of 25 g of xylitol in 50 mL water by ten healthy volunteers, the gastric emptying halftime was increased from 58 min to 91 min compared to the water only control as well as the 25 g glucose comparator in a crossover study. Food intake after xylitol preloading was reduced from 920 (water control) to 690 kcal [86]. Similar observations were made by King and co-workers [83] who observed that during a ten-day ingestion of yogurt containing 25 g of xylitol, 90 min prior to lunch, reduced the combined caloric intake by 11.9%. This difference did, however, not reach statistical significance compared to control.

### 7.4. Benefits of Xylitol on Metabolic Health

Xylitol, although having a similar sweetness as sucrose and glucose, has different molecular properties and thus does not lead to an increase in blood glucose or insulin levels [83]. Xylitol has a glycemic index of 7 ± 7 compared to a value of 100 for glucose; not surprisingly, the serum insulin and C-peptide responses to xylitol are negligible [87]. Carbohydrate and lipid oxidation were not observed to be influenced when eight healthy non-obese males consumed a single dose of 25 g xylitol after an overnight fast [87].

In animal models of type-2 diabetes (7 rats/group), induced through high-fructose feeding and injection of streptozotocin, administration of xylitol at 2.5%, 5% and 10% in drinking water during 4, respectively 5 weeks has been observed to improve serum insulin concentration at all tested xylitol concentration and glucose tolerance at 10% but not 2.5% and 5% xylitol [80,88]. In a study with 10 obese and 10 lean, non-diabetic volunteers; nasogastric administration of 50 g xylitol in 300 mL water after an 8 h fasting, resulted in a small but significant increase in serum glucose after administrations of xylitol compared with placebo. The authors hypothesized that this could be due to a decrease in plasma glucose over time after placebo intake rather than an increase in plasma glucose after xylitol intake [82]. However, the small increase is in line with earlier reports [87] and can be explained by the normal metabolism of absorbed xylitol to glucose by the liver [7].

In a non-diabetic non-high-fat diet rat model, total cholesterol and low-density lipoprotein (LDL)-cholesterol were significantly reduced (approximately 50% and 75%, respectively) after three weeks in the 10% xylitol drinking water group (6 animals) compared to water only control (5 animals) [89]. In a fructose-streptozotocin-induced type 2 diabetic rat model, 7 animals/group, were fed 0 (control), 2.5%, 5%, and 10% dietary xylitol for 4 weeks a dose-dependent reduction in serum cholesterol was observed. This was in particular driven by a dose-dependent reduction in LDL-cholesterol, where 10% xylitol reached a level lower than the non-diabetic control animals [80]. A similar trend has been reported for humans as well, but only with high doses (40–100 g/day) of xylitol [90].

In a high-fructose streptozotocin-induced, diabetes animal model (7 rats/group), administration of 10% xylitol in drinking water was not found to improve serum triglycerides after 5 weeks as compared to diabetic animals in the unsupplemented control group [88]. However, a fructose-streptozotocin-induced type 2 diabetic rat model, 7 animals/group, were fed 0 (control), 2.5%, 5%, and 10% dietary xylitol for 4 weeks observed a dose-dependent increase in serum triglycerides [80]. A differential lipidemic response between healthy and type 2 diabetic animal models and humans has been suggested [91].

### 7.5. Conclusions

While there is some indication for improved short-term weight loss in animal models, the long-term data in humans is inconclusive. There is some indication that xylitol may influence satiety hormones and gastric emptying in humans. Whether this translates into an effect on weight management remains to be determined. The benefit of xylitol on metabolic health; in addition to the benefit of the mere replacement of sucrose, remains to be determined in humans. Although there are indications for reduced LDL-cholesterol with xylitol consumption, this would need to be confirmed with lower dietary doses in humans as well as the effect of xylitol on serum triglycerides.

## 8. Discussion

The dental health benefits of xylitol are well established [3]. Here, we have highlighted that xylitol also has other potential health benefits, Figure 3. Many of these are related to oral-pharyngeal health. Changes in the respiratory microbiota are associated with positive effects on respiratory infections, sinusitis, and acute otitis media. Also, the immune function modulating effects of xylitol may contribute to the reduction in respiratory-related infections. Furthermore, topical or oral administration of xylitol seems to have anti-inflammatory effects on immune function and could be beneficial in controlling for example skin inflammation. As a non-digestible, non-absorbed, selectively fermentable carbohydrate, xylitol also exhibits the characteristics of prebiotics. Xylitol consumption is associated with changes in microbiota composition and metabolic activity, and influences bowel and immune function, and positively influences bone health. Being a low caloric sweetener, xylitol may contribute to weight management; but also by stimulating satiety and contributing to improved serum cholesterol levels. Finally, the topical application of xylitol is associated with improved skin moisture and improved skin barrier.

There are thus many opportunities for additional health benefits of xylitol. However, a limitation is that many of these novel health end-points are mainly based on *in vitro* and animal studies, and limited human intervention studies. This is helpful for the exploration of new health targets and for their mechanistic understanding. Furthermore, it should be observed that animal studies often used 6%–20% of xylitol in the diet, which obviously is beyond what is feasible for human consumption. There is, therefore, a rationale and especially a need to investigate the feasibility of these potential health benefits in humans.

The purpose of the current review was to focus on xylitol. However, it may be relevant to place this into the perspective of other sugar alcohols; without embarking on an in-depth review. In addition to 4 g/day xylitol, one month of 4 g/day sorbitol and to a lesser degree 4 g/day mannitol but not 3 g/day erythritol reduced tetracycline induced bone resorption in rats [64]. Inhaled mannitol may improve some lung functions in cystic fibrosis patients as indicated in a recent Cochrane review [92]. Some polyols, such e.g., lactitol [93] and sorbitol [94], have been suggested to have prebiotic potential. For improving bowel function, lactitol appears to be the sugar alcohol of choice [95]. Mannitol can work as an antioxidant and protect hyaluronic acid in the skin [96]. Lactitol has been reported to stimulate secretory IgA production [97]. Erythritol causes no increase in blood serum glucose level [82]. While sorbitol and erythritol have been shown to reduce glucose absorption from the intestine and improve muscular glucose absorption *ex vivo* [98,99,100]. Thus, while other sugar alcohols have multiple potential beneficial health effects, xylitol seems to be the more versatile or more investigated one.

## Figures and Tables

**Figure 1 nutrients-11-01813-f001:**
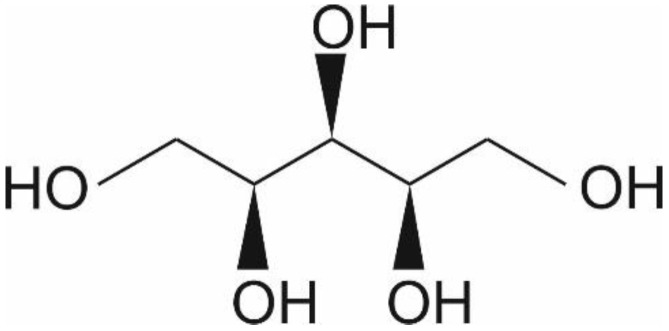
Chemical structure of xylitol ©DuPont Nutrition & Biosciences.

**Figure 2 nutrients-11-01813-f002:**
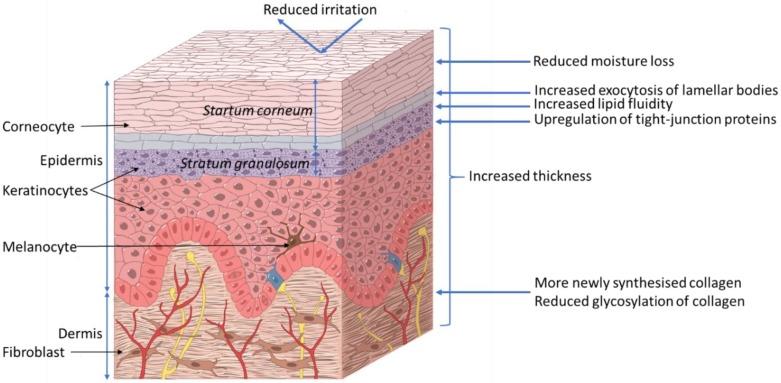
Proposed effects of xylitol on skin health. ©DuPont Nutrition & Biosciences.

**Figure 3 nutrients-11-01813-f003:**
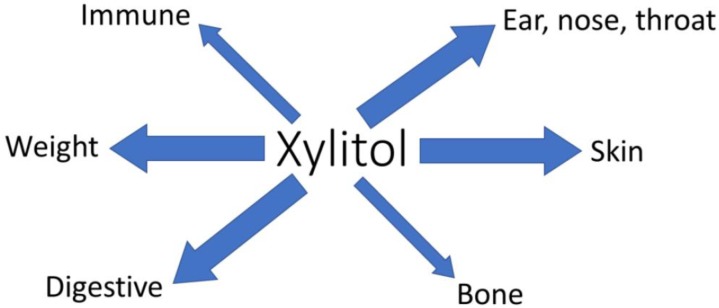
Summary of non-dental health benefits of xylitol. Arrow thickness indicates the level of documentation. Thin arrows indicate only *in vitro* or animal data, while thick arrows indicate some level of human data.

**Table 1 nutrients-11-01813-t001:** Non-exhaustive list of organisms that are able to grow or not to grow in the presence of xylitol, or that have the capacity to metabolize xylitol *in vitro* or not.

Organisms Reported to Grow on Xylitol	Reference	Organisms Reported Not to Grow on Xylitol	Reference
*Anaerostipes hadrus* (strain dependent), *A. caccae*	[27]	*Lactobacillus plantarum* 299v, *L. plantarum* 931, *L. rhamnosus* GG, *L. rhamnosus* LB21, *L. paracasei* F19, *L. reuteri* PTA5289	[35]
		*Bifidobacterium lactis* 1100, *B. lactis* Bb-12, *B. longum* 913, *B. lactis* 420, *L. acidophilus* NCFM, *L. casei* 921, *L. casei* Shirota, *L. bulgaricus* 365, *L. johnsonii* LA1, *L. paracasei* F19, *L. plantarum* 299v, *L. reuteri* SD2112, *L. rhamnosus* GG, *L. rhamnosus*, Lc-705, *Streptococcus mutans* Ingbritt	[36]
		*L. plantarum* 299v, *L. reuteri* DSM17938	[37]
		*Coprococcus catus, Eubacterium halli, E. limosum, E. rectale, Faecalibacterium prausnitzii, Megasphera elsedenii, Ruminococcus faecis, R. hominis, R. intestinalis, R. inulinivoruans*	[27]
		*S. pneumoniae*	
		*S. mutans, S. salivarius, S. sanguis*	
		*Candida albicans*, *S. mutans*	[38]
		*Staphylococcus epidermidis, Staphylococcus aureus, Pseudomonas aeruginosa*

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
