# Peer review of "Xylitol’s Health Benefits beyond Dental Health: A Comprehensive Review"

_nutrients, 2019, doi:10.3390/nu11081813_

Round 1
Reviewer 1 Report
The review authored by Krista Salli, Markus Lehtinen, Kirsti Tiihonen and Arthur C. Ouwehand provides an insightful summary of xylitol administration health benefits. Previously, xylitol was considered mostly in the context of dental health. Much smaller attention was attributed to other health areas covered by this review. Authors’ work is valuable as there is no comprehensive review on these topics. Moreover, the fact that the paper is based on current literature increases its scientific value. However, there are some issues that have to be improved.
Major issues:
1. The “Introduction” section is very limited for example in comparison to paragraphs in Section 2, where you have described skin morphology. Despite the fact that xylitol is a familiar substance to many researchers, the review should contain essential information to introduce the topic also for readers from other disciplines. Here you should provide information about chemical structure (and relation to other polyols), occurrence (products, supplements) and metabolism.
2. Similarly to Issue 1 – the fact of xylitol effectiveness in dentistry does not have to be known to readers. Please provide also a brief description of this area. Especially as the results are not coherent – a review from Cochrane Oral Health group indicated low quality of evidence for using xylitol for preventing dental caries.
3. The structure of review is not consistent – sections on the same organs/tracts are split: gastrointestinal tract into “probiotics” and “bowel function”, respiratory tract into “effects on the respiratory tract” and “sinusitis” or immune system into “Immune function” and “Anti-inflammatory”. Of course in the topic of gastrointestinal tract probiotics and motoric function of the bowel are separate issues but closely related to each other. But when you describe xylitol effects on respiratory health you refer to microorganisms occurring both in upper and lower tracts. Therefore for better readability, it will be helpful to reconsider these sections.
4. The influence of xylitol on microbiota is discussed in several sections on gastrointestinal, respiratory and immunological systems. This information can be described in one section on the influence of xylitol on microbiota (both physiological and pathological). Please describe more details on exact/proposed mechanisms of interaction between xylitol and microbiota.
5. You mention that “Anti-bacterial effects of xylitol have been well documented” – please extend this topic in the above-mentioned section.
6. Please extend the discussion with a comparison of xylitol health effects to other polyols.
7. Please include information about the safety of xylitol administration as well as potential adverse effects.
8. Data presented in your article are mostly based on animal and in vitro studies and any conclusions on health effects have to be made very carefully. Please discuss this important limitation.
Minor issues:
1. There is a number of text editing mistakes i.e. different line spacing or incoherent citation style
2. The figure about skin morphology is rather not needed as it is general knowledge. Instead, you should provide a comprehensive summary figure of health effects presented in your manuscript.
Reviewer 2 Report
the paper needs to be rewritten with more accuracy, specially regarding to clinical and preclinical studies. There are a lot, too many, scientific short-cuts. In-deep analysis of the papers you cited is required. None of the conclusion you give is properly demonsrated, in a scientific manner.
2. Skin
Line 56: be more accurate about the fucntions of filaggrin, loricrin, involucrin, occludins and about the model (keratinocyte in vitro, reconstituted skin …?)
Line 58: Same remark about the model + concentration of xylitol required for statistically significant effects + no effects on cell renewal ? epidermal differentiation = keratinocyte differentiation or not ? Keratinocyte is not the only cellular type in epididermis
Line 60: doses + how long the exposure ? + how many volunteers ? + placebo group or not ? which compound as control ? + how was dermal en epidermal thickness evaluated ?
Line 64: same remark. When glycerol was also applied, then could the effects really be attributes to xylitol ?
Line 74: which dose ? how long ? what kind of feed ?
Line 79-80: these finding suggest …diabetes … : no, this is a scientific shortcut and one of the mane mechanisms for DT2 is lipid accumulation resulting in sterile low grade inflammation. The results you present are not sufficient to support your hypothesis.
Line 84: be more accurate : dose of xylitol ? where ? how long ? in vitro or in vivo experiments ?
3. Prebiotic (3 + 4 together to have longer parts with similar length ?)
Line 89: where can xylitol be found in human food ? transporter ? other mechanism of absorption ? What is it becoming when absorbed ?
Line 95-98: I do not understand the sentence : what has been cultivated in vitro : colonic cells (which ones ?), small intestine cells (which ones) microbiota ? Was the concentration of butyric acid quantified ?
Line 102: an increase of what ? Has it been checked that there is no variation in the balance gram+/gram -, otherwise are the results are not relevant
Line 103: consumption of how much xylitol per day ? taken in only one meal or several ? how many volunteers ? control groups ?
Line 104: which yeast ? some are health benefic …. overall conclusion ? (103 to 110)
Lines 110-111: this needs to be suppress and formally demonstrated in the next paragraph
4. Bowel
Line 127: needs more accuracy and bibliographic reference(s)
5. Immune functions
Lines 153-155 : in vitro experiments ? or splenomegaly ? How many groups ? nothing, glucose alone (concentration), xylitol alone , B-mitogen + glucose vs B-mitogen +xylitol, T-mitogen + glucose vs T-mitogen +xylitol, how many animals per group ?
Line 171 : explanation ? trained immunity ? better innate immunity ? how ? Way of administration ?
6. Anti-inflammatory (These informations may go in part 2)
Lines 180 to 190 : is it really an anti-inflammatory effect ? rather a better barrier effect. As confirm lines 190 to 197
An anti-inflammatory effect would also be difficult to explain regarding to the previous paragraph: innate immunity needs inflammation.
7. Respiratory tract + 8 together
Line 164 to 178 are regarding to respiratory diseases
Lines 212 to 229: are they mandatory as at the end the results are not statistically significant ? Shorten ?
Line 242 and following : more accuracy as this is of interest : number of patients ? doses ? control or placebo groups ? …
9. Acute otitis OK
10. Weight management
Lines 282-90 : is it really the solution ? several studies demonstrated that sweetener do not allow long term weight reduction. You do not take into account the neurological aspect, with sweet taste inducing a pleasure/well-being response and reward pathway
10. Weight management and satiety
Line 295 : indicate what were the controls as solvent may also have an effect, even if it is water + which species : human, rat, mouse ? dose of xylitol ? when was it given regarding to food/feed consumption/ empty stomach/overnight without feed/food ?
Lines 303-304 : interesting => be more accurate in the description, specially control groups.
12. Metabolic health
Lines 314-15 : what was the control? Which type of diabetes ? how was it induced ? need for more accuracy.
Lines 315-18: same remark for accuracy, especially when was xylitol given (morning before breakfast, mid-morning …)?
Lines 319-27 : this is of interest only when the groups were pair-fed and isocaloric, otherwise it is not physiologically relevant
Conclusion of this paragraph: nothing has been proven .
12. Metabolic health
Lines 336, 341, 346: which model ? what did control groups receive ?
Lines 354-56 : the explanation is too short
Author Response
Please see attachement

Round 2
Reviewer 1 Report
The Authours generally improved their work and I think that the work is interesting and may be useful for other authors dealing with this research problem.
Author Response
The Authours generally improved their work and I think that the work is interesting and may be useful for other authors dealing with this research problem.
Response: Thank you.
Reviewer 2 Report
Efforts have been made but scientific shortcuts remain. There is a lack of rigor, exemple : wikipidia as a scientific reference !!
106 : what was the control ? answer = line 112 : conclusion : there is no scientific proof
117 : be lore accurate. How was the measure performed ?
120 to 127 : 1) be more accurate about amount given per day, how long, what was the feeding regimen ? control group received ? detail and describe the results with a critical point of view 2) streptozotocin = type 1 diabetes so you cannot say diabetes ..
137 : what did the control group receive ? 0.2% farnesol is the appropriate control, otherwise no conclusion can be drawn.
168 : what are relatively high doses ?
and so on ...
Round 3
Reviewer 2 Report
12: the papers do not allow you to say this
62: you have forgotten that in excess glucose can be stored as glycogen but also as triglycerides
100: was is the statistical value p<0.05 ? which test ? 7 volunteers but how many controls ? When no controls then it is not a valuable study.
105: statistically significant. When fold change is below 2, then it is not ? Protein quantitation was done as mRNA changes are not always correlated with proteins changes in the same extent. When not performed, then studies only suggest … and do not show
119: so you cannot say that it is xylitol that induces any of the changes you describe (meilleure hydratation)
124: what were the results when compared with 5% glycerol, that is a much better control than SDS alone.
128: the results do not suggest a polyol specific response, as the same effects are observed with glycerol …
129 word missing ? chow diet ? how many rats per groups ? how were statistical analysis performed
134: same remark : chow diet ?
136: type 1 diabetic male rats ? Were they also DT1 due to the same amount of streptozotocin?
139: word missing ?? similar observations on increased of what ??
140: chow diet ?
143: what is the interest to show that collagen is less glycosylated ?
177: reduced the growth of … ?
178: how were the statistical analysis performed ? Highest concentration = ?
180: when no comparison is done with 2% lactoferrin, then the study lacks proper control and is thus not valuable
184: same remark, the proper control should have been 0.2% farnesol
193: No, see previous remarks
196: the studies first need to be redone with proper controls before thinking to applicability to humans
209: be more accurate: not all short chain fatty acids contribute to support immune system homeostasis; the support is also mainly local. In addition, no short chain fatty acid supports alone the homeostasis of local immune system.
212: the increased concentration of xylitol in the digesta: to what concentration ?
215: no, this is not the definition of a prebiotic. So doing is there no growth of bacteria that are benefic to human health.
217: what was the composition of this colonic microbiota? Discussions are still ongoing to define what is a “normal” colon microbiota… How where the simulations performed: aerobic or anaerobic conditions? This needs to be far more accurate.
220: simulated colonic digesta ????
223: this does not mean that they are the one producing butyric acid …
296: may cause instead of caused as there is no proper statistical analysis, and what is a shift from Gram negative to Gram positive? More Gram + bacteria? but without any indication about the species involved, proper statistical analysis, this does not mean anything. As you suggest it in the following lines (300-302): eating glucose or xylitol has the same effect on distribution of bacteria according to Gram coloration.
307: what was the control ?
327: right bracket is not at the right place
384 : word “also” required ?
453: slightly reduced … either it is statistically significant, either it is not. + How were the statistical analysis performed (lines 450 to 453)?
457: what was the control (saline solution?)? and was there a control group ?
462: what was the xylitol concentration?
466: what is a trend ? It is significant or not.
474 : who demonstrated this ?
476 : very very partly explain this complex pathology !!
477 to 480: move to 6.2
493 to 498: this experiment tends to demonstrate that, whenever there is an immune modulation, it is very local, contrarily to what is written lines 477 to 480…
522: what is a moderate evidence? Statistically significant or not ?
554: what were the bones analyzed ?
568: the diets were also isolipidic? In amount and in quality (class of lipids, especially w3)
571: what was the control ?
604-605 : see my comments for skin
606: no, see my comments for oropharynx and your own conclusions
610: what is the link between immunity and the reference you mentioned?
613: intra-venous? intra-muscular? intra-peritoneal?
615: did the authors checked that it was not due to massive diffusion of xylitol in the body, contrarily to glucose, as its absorption is tightly regulated?
620: where were the activated neutrophils? In blood, in a tissue (which one/ones)?
621: is the difference statistically significant?
625: is the difference statistically significant?
626-27: see comments regarding to these sections
634: it indicates less acquired immune response, but not less or more immune response, as you said before xylitol to be able to improve neutrophils activation. Was his point checked in the study? In addition it is also possible that there is more CD3+CD4+ lymphocytes and consequently more antibodies, as you suggested earlier xylitol to improve antibodies production. Was this point checked? In any cases, this paragraph needs to be rewritten in a far more accurate and objective manner (lines 636-37).
640: what is remarkably reduced? What were the amounts of xylitol, red ginseng? What were the controls, the administration pathway: oral application = in water (and you are not sure of the amount really taken) or given as a single dose in water? What is the interest of red ginseng?
641: something goes wrong in this line (no verb in the sentence).
650: what is “in general effective”? Is there any percentage of the cases where it was ineffective? How many samples were analyzed? The sentence is not correct in steady state.
Jump in line numbers from 650 to 712 (but sentence is OK)
712-13: You first need to say that the authors demonstrate a significant increase of these cytokines with the TLR agonists, and next that the difference with xylitol is statistically significant.
713-14: TLR receptors for LPS (TLR4) and LTA (TLR2) are membrane ones, so there is no need for entrance into the cells or to cross the epithelial layer to activate them. Thus is your hypothesis not valuable for these 2 components.
717: what was the control? Lymphocytes in biopsies? Which lymphocytes? What about other immune cells, especially neutrophils, monocytes, macrophages, dendritic cells?
724-25: reduced physiological stress response: no, you do not demonstrate this.
726-30: I would s not say this, see comments regarding this section
739: the reference does not demonstrate what you write.
747, 748, 756: indicate p values
756: streptozotocin=DT1, where is fructose, in feed? How was diabetes checked ? what was the insulin production (glucose oral tolerance teste : should be more that untreated animals if DT2, no production if DT1)
758: I suppose the dose-dependent response is dependent on xylitol, I first understood it was on fructose.
758-60: Something goes wrong. I understand that the lowest dose of xylitol result in a dose-dependent decrease in feed and water intake, but that the highest dose does not. What are the control animals, the healthy animals? I think what you mean is that there is a statistically difference in food and water intake when HFD+xylitol groups are compared to the non-supplemented HFD group, with the highest concentration leading to the lower consumptions. This may result in less weight gain for HFD+xylitol groups than for the non-supplemented HFD group. What hypothesis did the authors do to explain this? In parallel, HFD+10%xylitol animals have similar weight gain than untreated animals???
Gap in line numbers 760 to 1216 but the, text is OK
1216: obese subjects: how were they defined as obese? Body-mass index?
1220: any information on their calories intake per day?
1222: obsess/lean: body mass index ?
1223: statistically significant? No comparison with another sugar?
1227: had no influence instead of not influence
1244 : how were they measured ? at what times ?
1247-48: same remark as before for streptozotocin and diabetes checking
1249: what do you mean? Higher baseline levels of insulin? Glucose tolerance test was performed on fasted (18h) animals?
1255-56: why not? But this is depending on what time the analysis were performed after xylitol intake, and this is not mentioned…
1257 = lean animals with chow diet and no diabetes
1259: this is of interest only when the animals drunk the same volumes of water…
1260: same remark as before for streptozotocin and diabetes checking
1263: a trend does not exist. The difference is statistically significant or not. Humans were lean? No diabetes? Age, sex match…?
1265: same remark as before for streptozotocin and diabetes checking
1267: this not surprising when the authors have not made so that animals drink the same volume of water
1408: this is also not surprising, see previous remark + part of xylitol is absorbed..
1411: the paper you cited as reference is on rats
1414: the long-term data in humans are not inconclusive, they conclude that xylitol has no effect on weight loss. This is not the same.
1418: the indications are for rats only, not for humans
1421-39: Figure3 and discussion need to be rewritten, as they are not in agreement with was has been demonstrated. This is over-interpretation.
Abstract also needs rewritting